# Synergistic Antibiofilm Effects of Chestnut and Linden Honey with Lavender Essential Oil Against Multidrug-Resistant Otitis Media Pathogens

**DOI:** 10.3390/antibiotics14020146

**Published:** 2025-02-02

**Authors:** Virág D. Ángyán, Viktória L. Balázs, Marianna Kocsis, Béla Kocsis, Györgyi Horváth, Ágnes Farkas, Lilla Nagy-Radványi

**Affiliations:** 1Department of Pharmacognosy, Faculty of Pharmacy, University of Pécs, 7624 Pécs, Hungary; angyan.virag@edu.pte.hu (V.D.Á.); viktoria.balazs@aok.pte.hu (V.L.B.); horvath.gyorgyi@gytk.pte.hu (G.H.); lilla.radvanyi@aok.pte.hu (L.N.-R.); 2Department of Agricultural Biology, Institute of Biology, University of Pécs, 7624 Pécs, Hungary; kocsis.marianna@pte.hu; 3Department of Medical Microbiology and Immunology, Medical School, University of Pécs, 7624 Pécs, Hungary; kocsis.bela@pte.hu

**Keywords:** chestnut honey, linden honey, lavender essential oil, antibacterial effect, biofilm eradication

## Abstract

Background/Objectives: Bacterial resistance to antibiotics is a major problem in healthcare, complicated by the ability of bacteria to form biofilms. Complementary therapy for infectious diseases can rely on natural substances with antibacterial activity, e.g., essential oils and honeys. The aim of the study was to investigate the effects of linden and chestnut honeys, lavender essential oil, and their combinations against the multidrug-resistant otitis media pathogens *Haemophilus influenzae*, *H. parainfluenzae*, *Moraxella catarrhalis*, *Pseudomonas aeruginosa*, and *Streptococcus pneumoniae*. The efficacy of these natural substances was compared with each other and antibiotics used in clinical practice. Methods: Microscopic pollen analysis and physicochemical traits were used to confirm the botanical origin of honey samples. The antibiotic sensitivity of bacteria was tested with a disk diffusion assay. Minimum inhibitory concentrations were determined using a microdilution assay. A 24 h immature biofilm eradication test was performed with a crystal violet assay. The efficacy of combinations was tested with a checkerboard titration method. The DNA release of damaged bacterial cells was measured using a membrane degradation assay. Results: Lavender essential oil displayed more potent antibacterial activity compared to the honey samples. However, honey–essential oil combinations showed higher inhibition rates for biofilm eradication, with *P. aeruginosa* being the most resistant bacterium. The combined use of chestnut honey and lavender oil resulted in a higher degree of membrane degradation in a shorter time, and their synergistic effect was proven with checkerboard titration. Conclusions: The combination of linden or chestnut honey with lavender essential oil was shown to be effective in the eradication of a 24 h immature biofilm formed by *H. parainfluenzae*, *M. catarrhalis,* and *S. pneumoniae*.

## 1. Introduction

Bacterial resistance to antibiotics appears in three main forms: intrinsic, acquired, and adaptive resistance. Through intrinsic resistance, the bacterium naturally resists the effects of certain antibiotics due to its innate structural and functional characteristics (the structure of the bacterial cell membrane, efflux pumps, and enzymatic processes, e.g., penicillin-degrading beta-lactamase) [1,2]. In case of acquired resistance, a bacterium previously sensitive to an antibiotic loses its sensitivity, which can develop through genomic mutations or horizontal gene transfer (HGT). During HGT, bacterial cells are able to transfer resistance-carrying genes to each other; thus, it is not necessary to develop resistance from generation to generation, which complicates treatment of the infection [3]. Adaptive resistance is a phenomenon in which bacteria are able to adapt to environmental stressors and the presence of an antibiotic without undergoing genetic change. In contrast to intrinsic and acquired resistance, adaptive resistance ceases when the trigger is removed [4].

Biofilm formation can be observed in many bacterial strains and has given bacteria a huge advantage, as bacterial biofilms provide optimal conditions and protection for the entire bacterial community, thus contributing greatly to the development of antibiotic resistance [5,6]. The dense polymer matrix of biofilms provides physical protection for the bacterium against the antibiotic; its cohesive structure mainly consists of extracellular polysaccharides, DNA fragments, and proteins [7]. Until recently, only biomaterial bound to the surface has been considered a biofilm, but, today, free-floating, aggregated bacterial cells are also recognized as biofilms [8]. Bacterial colonies that do not adhere to the surface have also been observed in children with persistent soft tissue infections or serous otitis media. In the latter case, a bacterial biofilm formed not only on the mucous membrane but also in the fluid of the middle ear [9,10].

Acute otitis media is the most common childhood illness and the most frequent reason for antibiotics to be prescribed in this age group to treat the infection [11]. It is a disease that occurs at least once in 50–85% of children by the age of three [12]. In many cases, it is a condition that develops after an upper respiratory viral infection, which can be accompanied by ear pain, hearing loss, fever, and the appearance of purulent ear discharge. As a result of the infection, the Eustachian tube loses its normal function, i.e., drainage of accumulated fluid, ventilation of the middle ear, and equalization of pressure. The accumulation of fluid caused by a blocked ear horn provides favorable conditions for the growth of bacteria [13]. The infection is often caused by biofilm-forming bacteria, which can reduce the success of antibiotic therapy, thereby slowing down the recovery process. The most common multidrug-resistant, biofilm-forming pathogens causing otitis media include Gram-negative *Haemophilus influenzae*, *H. parainfluenzae*, *Moraxella catarrhalis*, and *Pseudomonas aeruginosa* and Gram-positive *Streptococcus pneumoniae* [14]. Patients with acute otitis media have a high risk of the infection re-flaring, so it is very important to keep the ear canal clean and isolated and to promote the regeneration of the eardrum. Plant-based active ingredients with antibacterial activity (e.g., essential oil, honey) can play a significant role as a complementary treatment accompanying antibiotic therapies, as they can prevent the re-emergence of inflammation and superinfection.

Essential oils (EOs) are mainly composed of volatile hydrocarbons and oxygen-containing compounds, which are classified based on their biosynthetic pathways. Terpenoids are produced via the mevalonate pathway, while phenylpropanoids are synthesized through the shikimate pathway [15]. Many EOs penetrate cell membranes and, thanks to this, are able to influence the functioning of molecular targets [16]. Due to its anti-inflammatory, soothing, and antiseptic effects, lavender essential oil (Leo) can be used in dermatology to treat eczema and psoriasis. During upper respiratory infections, inhalation of Leo disinfects the airways [17]. Leo distilled during the early flowering period was proven to have a more significant anti-inflammatory effect compared to Leo distilled in the main flowering period or after flowering [18]. Its main components include the monoterpenes linalool, linalyl acetate, 1,8-cineole, terpinen-4-ol, β-ocimene, and camphor [19].

Honey is a natural medicinal substance that contains the active ingredients of the nectar source plant and the special enzymes and hormones of bees. The bactericidal hydrogen peroxide produced through the dilution of honey, the plant-derived polyphenols that increase the permeability of the bacterial cell membrane, the peptides synthesized by bees (e.g., defensin-1 peptide), the slightly acidic pH (bacteriostatic effect), and the high sugar content (osmotic stress) all contribute to honey’s antibacterial activity [20,21,22,23,24]. The strong antibacterial effect of linden honey (Lh) and chestnut honey (Ch) has been confirmed by several research groups, suggesting that their use as adjunctive therapy may also be effective in the case of otitis media [25,26,27,28,29,30]. In terms of polyphenols, lindenine was identified as a marker compound in Lhs, accompanied by higher amounts of ferulic acid and methyl syringate [31,32,33], while kynurenic acid content was prominent in Chs [34,35,36].

The therapeutic use and combination of medicinal substances that contain different types of active components are justified by their diverse and complex active ingredient composition [37]. Honeys and EOs might achieve their antibacterial effect through several modes of action, which in turn might hinder acquired bacterial resistance to these complex mixtures of compounds [38,39]. The basic question of our research is how effectively the combinations Leo–Lh or Leo–Ch can eradicate the biofilm formed by the most common multidrug-resistant otitis media bacterial strains in comparison to antibiotics and Leo, Lh, or Ch applied alone. To achieve these objectives, minimum inhibitory concentrations (MICs) were measured using microdilution assay, and biofilm eradication was investigated with crystal violet staining. To investigate the mode of action, a membrane degradation study was performed, and the level of interaction between honey and essential oil was determined through checkerboard titration.

## 2. Results

### 2.1. Pollen Profile and Physicochemical Parameters of Honey Samples

Based on the melissopalynological analysis, our Lh and Ch samples can clearly be regarded as varietal honeys, as they contained the pollen of the supposed nectar source plant in the percentage described in the literature (Table 1) [40]. In the case of Lh, linden (*Tilia* spp.) pollen (Figure 1A) was present in the largest proportion, but a significant part of pollen grains was identified as acacia (*Robinia pseudoacacia*) and oilseed rape (*Brassica napus*) pollen. Complying with requirements regarding Ch, chestnut (*Castanea sativa*) pollen (Figure 1B) was present in more than 90% in our honey sample. The sensory and physicochemical properties of the honey samples (Table 2) also supported their monofloral origin, in accordance with the results of the microscopic pollen analysis.

### 2.2. Antibiotic Sensitivity of Test Bacteria

Table 3 and Appendix A summarize the resistance profile of the pathogens included in the study. Based on these results, amikacin was used for *H. influenzae* and *H. parainfluenzae* bacteria, imipenem for *M. catarrhalis* and *S. pneumoniae*, and gentamicin for *P. aeruginosa*.

### 2.3. Minimum Inhibitory Concentrations (MICs)

The MIC values of the studied samples (Lh, Ch, and Leo) and the antibiotic controls are summarized in Table 4. The most resistant bacterium was *P. aeruginosa* in the case of both the honey samples and the essential oil, while Gram-negative *Haemophilus* species and Gram-positive *S. pneumoniae* bacteria were the most sensitive in the case of honeys and essential oil, respectively. Antibiotics were more effective against each bacterial strain included in the study compared to Lh, Ch, and Leo. Comparing the efficacy of honeys and Leo, the antibacterial activity of Leo was more significant. The Tween40 used for the emulsification of Leo had no antibacterial effect.

### 2.4. Biofilm Eradication Activity

In this experiment, Lh, Ch, Leo, and the antibiotics were used at MIC/2 concentration when tested alone, but in the case of honey–essential oil combinations, MIC/4 values were used, as our goal was to reveal the effect of the combinations on biofilm eradication, not to destroy the bacterial cells. Based on our observations, all samples had antibiofilm activity, but they had a different effect on the biofilm eradication of the five bacterial strains included in the study (Figure 2). The biofilm formation of *H. influenzae* (Figure 2A) and *H. parainfluenzae* (Figure 2B) was inhibited to a greater degree by the honey samples, with Lh displaying higher inhibition rates compared to Leo. In the case of *M. catarrhalis* (Figure 2C), however, the essential oil had a higher inhibition rate compared to the honey samples, and here Ch was the stronger inhibitor from the honey samples. Similar tendencies were observed in the case of *P. aeruginosa* (Figure 2D) and *S. pneumoniae* (Figure 2E) bacteria, where Ch and Leo exerted the same level of inhibition, while the inhibition activity of Lh was somewhat lower. For each bacterial strain, the combinations had the strongest antibiofilm activity (Ch + Leo inhibition rate: 74–93%). The antibiotic had the lowest inhibitory effect on biofilm formation (inhibition rate: 28–34%). Gram-negative *P. aeruginosa* was the most resistant, while the Gram-positive *S. penumoniae* bacterium was the most sensitive. Biofilm formation was not affected by the emulsifier (Tween40) used in the case of essential oil.

### 2.5. Checkerboard Assay

The results of checkerboard titration are summarized in Table 5 and Table 6. Table 5 shows the combination MIC values of the test substances, which were used to calculate the FICI values. Based on the FICI values, the interactions between the test substances were determined, taking into account the following value ranges: FICI ≤ 0.5: synergist; 0.5–1: additive; >1: negative interaction; >4: antagonist [41].

Based on the FICI values, Ch used together with Leo resulted in a synergistic effect against both Gram-negative and Gram-positive bacteria. When Leo was combined with Lh, an additive effect was detected (Table 6).

### 2.6. Membrane Degradation Assay

The checkerboard titration supported the synergistic effect of Ch and Leo (Table 6), so membrane degradation studies were performed with these two samples and their combinations. In addition, Ch was chosen for these experiments because of its high antibacterial and antibiofilm activity against *S. pneumoniae* (Gram-positive) and *P. aeruginosa* (Gram-negative) bacteria (Table 4, Figure 2).

Using Ch and Leo separately at MIC × 2 concentrations, bacterial DNA release was above 60% (60.7–68.4%) for both bacterial strains included in the study. In the case of the combination, half of the concentration of the two natural substances (MIC, 1:1 ratio) resulted in DNA leakage of 67.9% (*P. aeruginosa*) and 69.8% (*S. pneumoniae*) (Table 7). In the study of DNA release kinetics, Ch and Leo were used separately at MIC × 2 concentrations at different time intervals (20, 40, 60, and 90 min) (Table 8). For the combination, in this case too, we worked with the MIC concentration for both samples (1:1 ratio). In each measurement, bacterial DNA was detected already at 20 min (34.4–53.6%), and the highest values were obtained at 90 min (66.6–71.2%). In this study, as well, the most DNA was released during treatment with the combination (90th minute: 70.9 and 71.2%). Based on our results, *P. aeruginosa* proved to be a more resistant bacterium compared to *S. pneumoniae* (Table 7 and Table 8).

## 3. Discussion

Investigating the antibacterial and biofilm inhibition activities of natural substances is vital to explore new therapeutic options to curb the increasing spread of antibiotic resistance. In our study, we applied a novel approach by exploring the possibility of using two natural substances, honey and essential oil, together. Chestnut and linden honey, which were included in our experiment, have proven antibacterial and biofilm formation inhibitory effects against respiratory bacteria. Combining them with Leo not only resulted in an increased antibacterial effect, but the anti-inflammatory effect of lavender essential oil can also contribute to the success of the complementary therapy. Both honey and essential oil have been hypothesized to achieve their antibacterial efficacy through several modes of action.

Essential oils are multicomponent, lipophilic liquids with characteristic aroma, as they comprise volatile compounds. Up to 20–60 different bioactive components can be present in essential oils [42]. Their mechanisms of action are diverse; some essential oils can kill bacteria (bactericidal effect), but there are some that only slow down their division (bacteriostatic effect) [43]. The general bacteriostatic effect of essential oils is based on several mechanisms. They cause the disintegration of the bacterial outer membrane or the phospholipid bilayer. They increase the permeability of the bacterial cell membrane and promote the leakage of intracellular components from the cell, which leads to an upset in the ionic balance. In addition, they can interfere with the cellular metabolism (glucose and fatty acid metabolism) and enzyme kinetics of pathogens [44,45]. According to some researchers, there is a significant difference between the sensitivity of Gram-negative and Gram-positive bacteria. Essential oil components with hydrophobic properties are more effective on Gram-positive bacteria [46,47]. This phenomenon can be explained by the fact that Gram-positive bacteria have a thick peptidoglycan layer linked to apolar molecules through which lipophilic components can easily pass; meanwhile, in the cell envelope of Gram-negative bacteria, proteins linked to the outer membrane and lipopolysaccharides prevent this [48]. Understanding the mechanism of action of a given essential oil or one of its components is of paramount importance, as this allows us to develop new therapeutic procedures and identify new areas of indication. The antibacterial effect of lavender is well-known, and numerous studies have supported the antibacterial and anti-inflammatory effects of Leo [18,49,50]. However, there are few studies on the antibacterial and anti-biofilm effects of Leo against pathogens that cause upper respiratory tract diseases. The activity of Leo against *Haemophilus* strains was first described by our research group. Due to the overuse of antibiotics and the increasing phenomenon of antibiotic resistance, which can largely be attributed to the ability of bacteria to form biofilms, research is directed towards bacteria that cause problems in the nosocomial area [51,52,53]. Leo has been shown to be effective against both MRSA (methicillin-resistant *Staphylococcus aureus*) and VRE (vancomycin-resistant *Enterococcus faecalis*) at concentrations below 1% [54]. In the study by de Rapper, the efficacy of Leo was tested in combination with antibiotics and antifungals, and the individual MIC value of Leo was also determined using the microdilution method. The MIC values obtained against the bacteria *P. aeruginosa* and *S. aureus* were 1 mg/mL, and values against the fungus (*C. albicans*) were 3 mg/mL. Another research group measured 2.33 mg/mL as the MIC value of Leo against the pathogen *P. aeruginosa* [55]. The MIC value of Leo against *P. aeruginosa* in our study was 2.5 mg/mL, which shows that our data are comparable with the results of other research groups [56]. A veterinary study indicated that Leo may be effective as a topical treatment for canine otitis externa promoted by biofilm-producing *P. aeruginosa* [57]. The antipseudomonal effect of *L. angustifolia*, *L. intermedia*, *L. pyrenaica,* and *L. stoechas* essential oils was confirmed by Végh et al. (2012) [58]. The anti-*Pseudomonas* activity of essential oils obtained from *L. sumian* and *L. grosso* was supported by Donadu’s research group. Their results confirmed that *L. sumian* essential oil showed lower antimicrobial activity (MIC: 16% solution) than *L. grosso* (MIC: 8% solution). None of the Leos tested showed cytotoxic effects at the tested concentrations [59].

In biofilm inhibition studies, *P. aeruginosa* was the most resistant pathogen in each test system. This may be due to the fact that this bacterium has three exopolysaccharides: Psl, Pel, and alginate. Psl is a neutral pentasaccharide that typically contains D-glucose, D-mannose, and L-rhamnose moieties. Psl acts as a signaling molecule and promotes the production of c-di-GMP (bis-(3′-5′)-cyclic dimeric guanosine monophosphate), the levels of which, when elevated, result in thicker and more robust biofilms [60]. In addition, Psl protects biofilm bacteria from antimicrobial agents and neutrophil phagocytosis [61]. Pel is a cationic polysaccharide polymer of partially deacetylated N-acetyl-D-glucosamine and N-acetyl-D-galactosamine. Pel promotes the tolerance of bacteria embedded in biofilms to aminoglycoside antibiotics [62]. Alginic acids are mixtures of uronic acid polymers, the main components of which are β-D-mannuronic acid and α-L-guluronic acid. The salts of alginic acids are alginates. Alginate has a wide range of functions, including biofilm maturation, protection against phagocytosis, and reduced diffusion of antibiotics through the biofilm [63]. The aforementioned units contribute to the increased resistance of *P. aeruginosa* biofilm to treatment. The other pathogens included in the study (*Haemophilus* spp., *M. catarrhalis*, *S. pneumoniae*) do not have such resistant biofilms, so higher inhibition rate values were detected for these pathogens.

Among the possible mechanisms of action underlying the effectiveness of Leo, in addition to membrane-degrading activity, the inhibition of quorum sensing (QS) should also be mentioned. Although we do not have enough data regarding the anti-QS effect of Leo, a study written in 2014 supports this fact [64]. The antibacterial effect of vaporizing Leo was also confirmed against *Staphylococcus* strains (*S. hominis*, *S. haemolyticus*, *S. epidermidis*, *S. aureus*) in hospital environments. The data showed that Leo reduced the number of bacteria in all hospital areas compared to the untreated control [65].

Based on the microbiological studies conducted to date, it can be concluded that similar studies related to *Haemophilus* strains and *M. catharralis* and *S. pneumoniae* bacteria are not available. Furthermore, it should be emphasized that few studies investigate the biofilm inhibitory effect of Leo [66]. In addition to the above, another novelty of our work is that we also examined the biofilm eradication activity of Leo in combination with two types of honey. Honeys, just like essential oils, are promising antibacterial and antiviral agents due to their diverse plant-derived compounds and multiple modes of action.

The antimicrobial activities of honey have recently been reviewed by Luca et al. (2024) [67]. Honey’s antibacterial activity was reported against numerous bacteria, including *Acinetobacter*, *Escherichia*, *Klebsiella*, *Pseudomonas*, *Staphylococcus,* and *Streptococccus* strains. The antifungal effect of honey was investigated mainly against various *Candida* and *Aspergillus* species. The antiviral activity of honeys requires further research; however, honey seems to be active against SARS-CoV-2, human immunodeficiency virus, herpes simplex virus types 1 and 2, viral hepatitis virus, and others [68,69]. The antimicrobial properties of honey are attributed partly to plant-derived compounds, like polyphenols [23], and partly to bee-derived components, like defensin-1 peptide [20]. The antibacterial activity of honey is usually in correlation with its hydrogen peroxide content [70], e.g., in acacia, rapeseed, and sunflower honeys harvested in Europe but not in linden honey. The synergistic effect of polyphenols and hydrogen peroxide has been proposed to determine a honey’s antibacterial qualities [71]. New Zealand’s manuka honey is characterized by non-peroxide antibacterial activity, which can be attributed to its methylglyoxal content [72]. In addition, the physicochemical traits of honeys, like acidity (low pH), high sugar concentration, low water content, phenolic acids and flavonoids, as well as proteins, all contribute to honey’s antibacterial properties. It must be emphasized that individually none of these factors would be sufficient to achieve antimicrobial activity, as their complex interaction is required [67]. Exposure to honey can induce physiological changes in bacteria related to membrane integrity and polarization and can lead to metabolic disruption [29]. High levels of carbohydrates in honey can cause the dehydration of bacterial cells through osmosis [67]. Damage to the cytoplasmic membrane can result in the easy release of nucleic acids, amino acids, K+, proteins, and inorganic phosphates from the bacterial cell, leading to cell lysis and death [73]. Lh and Ch contain higher proportions of the polyphenolic components ferulic acid and gallic acid, which can cause irreversible damage to membrane properties of various pathogenic bacteria (e.g., *P. aeruginosa*), such as physicochemical characteristics or intra- and extracellular permeability [74,75,76]. Increased pore formation, local tearing, or a decrease in negative surface charge may occur [23,77]. Membrane-bound efflux pumps (proteins) fulfill many functions in bacterial cells; for example, they promote the development of resistance to antibacterial agents, biofilm stabilization, and protection against oxidative stress [78,79,80]. Efflux pump inhibitors are used to prevent antibiotic resistance of multidrug-resistant bacterial strains with increased efflux activity. These compounds significantly reduce tolerance and biofilm formation and thus increase the effectiveness of antibiotics [81,82]. Phenolic compounds found in honey (e.g., ferulic, gallic, caffeic and salicylic acids, quercetin, kaempferol, galangin) can act as efflux pump inhibitors by inhibiting ATPases involved in antibiotic efflux [79,83]. In order to increase their chances of survival, bacterial cells constantly communicate with each other using signal molecules. It has been observed that the QS mechanism (local cell density sensing) of bacteria directly affects the development of antibiotic resistance through the expression of different efflux pumps [84]. QS activity is significantly inhibited by many unifloral honeys (e.g., by degrading the N-acyl homoserine lactone signaling molecule in Gram-negative bacteria); however, the extent of the inhibitory effect depends significantly on the type of honey, the place of collection, or the different phenolic acid content. For example, chestnut and linden honeys show higher QS inhibitory activity than orange or rosemary honeys [79,85,86].

There are a few studies available on the combination of honey and essential oil. Imtara and his research group used checkerboard titration to explore the possible synergistic effect of oregano (*Origanum vulgare*) essential oil and different honey samples on *E. coli*, *P. aeruginosa*, *S. faecalis,* and *Staphylococcus aureus* by calculating fractional inhibitory concentrations. The most pronounced synergistic effect was achieved with the combination of oregano essential oil and *Euphorbia* honey against the pathogens *E. coli* and *Streptococcus faecalis* [87]. Another study supports the synergistic effect of combining eucalyptus honey and eucalyptus essential oil against the pathogens *E. coli*, *Proteus mirabilis*, *Salmonella* Typhimurium, *Bacillus subtilis*, *Staphylococcus aureus*, *Listeria monocytogenes*, *Candida albicans*, *Trichophyton rubrum,* and *Aspergillus niger*. The study tested honey and essential oil separately and in combination. The results confirmed that the combination of the two substances was more effective [37]. The research team of Qasem tested the combination of chamomile essential oil and chamomile honey against similar pathogens. The results showed that the most sensitive pathogens were *S. aureus* and *L. monocytogenes*, and the most resistant ones were *E. coli*, *P. mirabilis*, *S.* Typhimurium, and *P. aeruginosa*. In all cases, the combination of chamomile honey and chamomile essential oil showed higher efficacy compared to the use of honey or essential oil separately [88]. Increased efficacy of honey–essential oil combinations can be attributed to the fact that complexity makes it difficult for bacteria to develop tolerance, as adapting to several different active ingredients is a much more time-consuming and complicated task.

Our study demonstrated the synergistic effect of Ch and Leo against otitis media pathogens through a series of in vitro experiments. Using only in vitro assays poses a limitation for the usability of our data; thus, further experiments should be planned to test the honey–essential oil combinations in either in vivo or clinical settings. Further research should be directed to clarifying which constituents of honeys and essential oils are responsible for their respective antibacterial and antibiofilm effect and how these bioactive compounds can enhance each other’s efficacy in combination, resulting in synergistic effects. The current study revealed the membrane degrading potential of Lh, Ch, and Leo. This is, however, only one of the mechanisms of honeys’ and essential oils’ antibacterial action; thus, further experiments are needed to shed light on whether these complex natural substances or their bioactive compounds are able to inhibit other defense lines of bacteria, such as efflux pumps and quorum-sensing mechanisms. In addition, the study could be broadened to include several other bacterial strains causing multiple types of infections to reveal the width of the antibacterial spectrum of honeys and essential oils.

## 4. Materials and Methods

### 4.1. Honey and Essential Oil Samples

For our investigations, linden (*Tilia* spp.) and chestnut (*Castanea sativa* Mill.) honeys were purchased in 2021 from Hungarian apiaries in the South Transdanubian region; Lh was purchased from the Reith Apiary (Baranya County) and Ch was purchased from the Németh Apiary (Zala County). The aerial parts of lavender (*Lavandula angustifolia* Mill.) were collected before the flowering period in Bolhó village (N46.0408 E17.3034, Somogy County, Hungary) in 2021. The plant material was prepared for essential oil extraction, as described in our previous publication [18]. The herba was dried at room temperature for 1 week, and then the essential oil was extracted through hydrodistillation [89]. The lavender essential oil (Leo) distilled in the pre-bloom period was selected for this study based on our previous experiments, where Leo from this phenophase exhibited greater antibacterial and anti-inflammatory activity compared to Leo obtained during and after the flowering period [18,90]. The chemical composition of Leo was characterized in Balázs et al. (2025), with the main components being linalool (31.02%) and linalyl acetate (23.66%), followed by terpinen-4-ol (6.44%), α-terpineol (5.54%), and geranyl acetate (3.45%) [90].

### 4.2. Melissopalynological Analysis

Following the work of Nagy-Radványi et al. (2024—A) [91], we chose microscopic pollen analysis as one of the most effective methods to determine the botanical origin of honeys. Honey pollen samples were prepared according to the methodology developed by Von der Ohe et al. (2004) [92]. Ten grams of honey was measured into a 50 mL centrifuge tube and then mixed with 20 mL of distilled water using a Combi-spin FVL-2400N vortex (Biocenter Ltd., Szeged, Hungary). The centrifuge tube was placed into a Neofuge 15R centrifuge (Lab-Ex Ltd., Budapest, Hungary) and centrifuged at 3000 rpm for 10 min. After removing the supernatant, the remaining sediment was supplemented with 10 mL of distilled water, and the mixture was centrifuged again at 3000 rpm for 5 min. After the second centrifugation, 250 µL of distilled water was added to the sediment remaining in the centrifuge tube and vortexed, and then 20 µL of this pollen suspension was pipetted onto slides on an OTS 40 heating plate (Tiba Ltd., Győr, Hungary) set to 40 °C. A piece of Kaiser’s glycerin jelly with fuchsine (Merck Life Science Ltd., Budapest, Hungary) was placed onto the pollen preparations, thus staining the pollen grains. The pollen preparations were covered with a coverslip and analyzed with a Nikon Eclipse E200 light microscope connected to a Michrome 20 MP CMOS camera (Auro-Science Consulting Ltd., Budapest, Hungary) at 400× magnification using the 4.3.0.605 version of TCapture software [93]. Five hundred pollen grains were counted for each honey sample, and the plant species or family the particular pollen grain belonged to was recorded. The relative frequency of pollen types was given as a percentage of all pollen grains.

### 4.3. Physicochemical Parameters of Honey

The physicochemical properties of the honey samples were determined based on the research of Nagy-Radványi et al. (2024—A) [91]. For the determination of pH and electrical conductivity, we used a DSZ-708 multiparameter analyzer (Simex Ltd., Budapest, Hungary). The pH of the honey samples was measured according to the Hungarian Food Codex, and the pH meter was calibrated using buffer solutions with pH values of 4 and 9 [94]. For the determination of electrical conductivity, 20% (*w*/*w*) aqueous honey solutions were prepared, while for pH measurement, 10 g of honey was dissolved in 75 mL of carbon-dioxide-free distilled water. The electrical conductivity of honey, which is related to its mineral content, was measured at 20 °C, and the results were expressed in milli-Siemens per centimeter (mS/cm) [95].

To analyze the color intensity of varietal honeys, 50% (*w*/*w*) aqueous honey solutions were prepared, placed in an ultrasonic water bath (45–50 °C) for 5 min, and then filtered (0.45 µm pore size, Agilent Technologies, Milan, Italy). Spectrophotometric analysis was carried out using a Shimadzu UV-1800 spectrophotometer (Shimadzu Schweiz GmbH, Reinach, Switzerland). The results, expressed in milliabsorbance units (mAU), were calculated from the difference in absorbance values measured at 450 nm and 720 nm [96].

### 4.4. Bacterial Strains

In our research, multidrug-resistant bacteria that play a significant role in the development of otitis media were used. The pathogens *Moraxella catarrhalis* (DSM 9143), *Pseudomonas aeruginosa* (ATCC 27853), and *Streptococcus pneumoniae* (DSM 20566) were cultured using Brain Heart Infusion (BHI) Broth (Sigma-Aldrich Ltd., Budapest, Hungary) as the liquid medium. For culturing two members of the *Haemophilus* genus, *H. influenzae* (DSM 4690) and *H. parainfluenzae* (DSM 8978), a specialized growth medium was prepared. Specifically, 3750 µL of Mueller-Hinton II Broth (MHB, OXOID Ltd., London, UK) was supplemented with 750 µL of NAD solution (1 mg/mL) and 500 µL of *Haemophilus* Supplement B (Diagon Ltd., Budapest, Hungary). The bacterial cultures were incubated in a shaking incubator (at 60 rpm) at 37 °C for 12 h using a C25 Incubator Shaker (New Brunswick Scientific, Edison, NJ, USA) [97].

### 4.5. Determination of Antibiotic Sensitivity of Test Bacteria

The antibiotic sensitivity of the microorganisms was evaluated using the Kirby–Bauer disk diffusion method, based on the work of Nagy-Radványi et al. (2024—B) [98] and following the guidelines of the Clinical and Laboratory Standards Institute (CLSI) and the Manual of Clinical Microbiology [99,100]. Susceptibility to the following antibiotics (OXOID Ltd.) was examined: amikacin (30 µg), ampicillin (10 µg), amoxicillin/clavulanic acid (20/10 µg), ceftazidime (10 µg), ciprofloxacin (5 µg), erythromycin (15 µg), gentamicin (30 µg), imipenem (10 µg), levofloxacin (5 µg), oxacillin (1 µg), penicillin (1 µg), piperacillin–tazobactam (100/10 µg), and vancomycin (5 µg) [101]. A filter paper disk impregnated with the specified antibiotic concentration was placed onto the surface of the culture medium (in the case of *P. aeruginosa*, *M. catarrhalis* and *S. pneumoniae*: Mueller-Hinton Agar, Sigma-Aldrich Ltd., Budapest, Hungary; in case of *Haemophilus* spp.: chocolate agar) that had been inoculated with a bacterial suspension (equal to the visual turbidity of a no. 0.5 McFarland standard). After 15 min, the Petri dishes were incubated at 35 ± 2 °C for 16–18 h, after which the inhibition zones were measured under visible light.

### 4.6. Determination of Minimum Inhibitory Concentration (MIC)

To determine the MIC, the microdilution method was used in accordance with the guidelines of the CLSI [102]. The procedure was carried out on 96-well microtiter plates, and, to ensure accuracy, all tests were performed in six replicates. For linden and chestnut honey, the following dilution series was prepared using BHI broth: 22.5, 20, 17.5, 15, 12.5, and 10% (*w*/*w*). To emulsify the essential oil, a Tween40-containing medium was used, and the following dilution series was prepared: 10, 5, 2.5, 1.25, 0.625, and 0.3125 mg/mL. To each well, 100 µL of a 10^5^ CFU/mL bacterial suspension prepared in the appropriate growth medium was added, followed by 100 µL of honey solution or essential oil solution at varying concentrations. During our study, an antibiotic specific to each bacterial strain was used as the positive control. In the case of *Haemophilus* spp., amikacin (Likacin 250 mg/mL solution for injection (Lisapharma S.P.A., Erba, Italy), in the case of *M. catharralis* and *S. pneumoniae*, imipenem (Imipenem/Cilastatin Kabi 500 mg/500 mg powder for solution for infusion), and in the case of *P. aeruginosa*, gentamicin (Gentamicin Sandoz 80 mg injection, Sandoz) antibiotics were used. Bacterial cells were also treated with BHI solution containing 1% Tween40 without the addition of essential oil in order to exclude the effect of Tween40. The plates were incubated for 24 h at 37 °C, and then absorbance values were measured at 600 nm using a microplate reader (BMG Labtech, Budapest, Hungary) [98]. We considered the lowest honey or essential oil concentration as the MIC value, in which case no bacterial growth was visible in the wells of the microtiter plate.

### 4.7. Biofilm Eradication Assay

In order to investigate the effect of honey samples and Leo on biofilm eradication, biofilms were formed on 96-cell microtiter plates, similarly to Balázs et al. (2023) [103]. In these experiments, MIC/2 concentrations determined in previous MIC studies were applied for honey and essential oil samples, while MIC/4 concentrations were used for combinations.

To initiate biofilm formation, 200 µL of bacterial suspension at 10^8^ CFU/mL was pipetted into individual wells of the microtiter plates. As a negative control, BHI medium was applied, while untreated bacterial suspensions (10^8^ CFU/mL) served as the positive control. During a 24 h incubation period (37 °C), bacterial cells formed adherent and cohesive biofilms (24 h immature biofilm). Unattached cells were washed out with physiological saline, followed by the treatment step. Treatments involved adding 200 µL of honey or essential oil solutions or 100 µL of each solution for the combination (MIC/4 concentrations). Tween40-containing medium was used to emulsify the essential oil. After incubation (24 h at 37 °C), the wells were washed again with physiological saline to remove non-adherent cells, and the adhered cells were fixed with 99% methanol (Molar Chemicals Ltd., Halásztelek, Hungary) for 15 min at room temperature (23 °C). After removing the methanol, the biofilms were stained with 0.1% crystal violet for 20 min. The crystal violet binds to negatively charged molecules in the extracellular matrix of biofilms, enabling the quantitative determination of total biofilm biomass. As a final step, the dye bound to the biofilm was dissolved in 33% acetic acid. Absorbance was measured at 595 nm using a SPECTROstar Nano microplate reader (BMG Labtech, Budapest, Hungary) [104,105]. In the series of experiments, the following antibiotic controls were used: *H. influenzae* and *H. parainfluenzae*—amikacin, *M. catarrhalis* and *S. pneumoniae*—imipenem, and *P. aeruginosa*—gentamicin.

The inhibition rate was calculated using the formula [106]Inhibition rate = (1 − S/C) × 100%
where C is the absorbance of the control and S is the absorbance of the sample.

### 4.8. Checkerboard Assay

The combination studies were performed using the checkerboard titration method. This method is particularly suitable for studying the combination effect of substances with antibacterial effects. The selected bacterial strains were *P. aeruginosa* and *S. pneumoniae*; thus, the interaction between different substances was tested in the case of both Gram-negative and Gram-positive pathogens. BHI medium (Sigma-Aldrich Ltd.) was used for cultivation; the germ count was 10^5^ CFU/mL. Furthermore, 96-well microtiter plates were used for the studies, and 3 parallel measurements were performed. In this study, Leo was combined with Lh and Ch for both pathogens. In each study design, a halving dilution series was prepared at concentrations corresponding to predetermined MIC values, which were as follows: MIC × 2, MIC, MIC/2, MIC/4, and MIC/8. Subsequently, according to the test layouts, 100 μL of bacterial suspension and then 50 μL of essential oil sample (substance A) and 50 μL of honey sample (substance B) were applied to the cells of the microtiter plates. In parallel, a bacterial suspension without samples was applied to a microtiter plate as the positive control, and only BHI medium was used as the negative control. The prepared microtiter plates were incubated (for 24 h at 37 °C), and then their absorbance was measured at a wavelength of 600 nm with a plate reader (BMG Labtech, Budapest, Hungary; Bio-Tek Ltd., Winooski, Vermont, USA). Following the test, combination MIC values were calculated. The concentration at which the absorbance of the positive control was reduced to ±10% was considered the combination MIC value. From the combination MIC values calculated in this way, the fractional inhibitory concentration index (FICI) was calculated, which is a dimensionless numerical value that helps to evaluate the combination effect of pharmacons.$$\mathrm{FICI}=\mathrm{FICA}+\mathrm{FICB}=\frac{MIC\;A\;combination}{MIC\;A}+\frac{MIC\;B\;combination}{MIC\;B}$$

### 4.9. Membrane Degradation Assay

During the study of membrane degradation, bacterial suspensions (10^8^ CFU/mL) were prepared in phosphate buffer solution (PBS), and untreated bacterial cells were used as controls. Samples (Ch, Leo, and honey—essential oil combination) with concentrations of MIC/4, MIC/2, MIC, MIC × 2, and MIC × 4 were added to the cells (incubation time: 1 h). The time dependence of DNA release was also investigated; in this test, the bacterial cells were suspended in PBS containing honey or essential oil with a concentration of MIC × 2. In the case of the combination, Ch and Leo were used in MIC concentration (1:1 ratio). Treatments lasted 0, 20, 40, 60, and 90 min. In both cases, these steps were followed by centrifugation (Neofuge 15R, Lab-Ex Ltd., Budapest, Hungary) at 12,000× *g* for 2 min. The absorbance of the supernatant liquid containing nucleic acid was determined at 260 nm with a Metertech SP-8001 (Abl&e-Jasco Ltd., Budapest, Hungary) spectrophotometer, and the results were expressed as a percentage compared to the control [107].

### 4.10. Statistical Analysis

Statistical analyses were carried out using Microsoft Excel^®^ 2016 MSO (16.0.4266.1001 version) (Microsoft Corp., Redmond, WA, USA) and the PAST software package version 3.11 [108]. The effects of honey types, essential oil, and their combinations were compared with each other using one-way ANOVA. If the null hypothesis of the ANOVA was rejected, we used Tukey’s test to establish a difference between the given groups (*p* < 0.05).

## 5. Conclusions

Based on our results and a few other studies conducted so far, it can be stated that the combined use of essential oil and honey may be promising for suppressing pathogens and eradicating biofilms. Demonstrating the synergistic effect of Ch and Leo supports that combinations of the two natural substances could be used in adjunctive therapy for otitis media. Our in vitro experiments provide a good starting point for the development of an ear drop or ear spray with the honey–essential oil combination that would speed up the healing time by preventing the infection from becoming chronic.

## Figures and Tables

**Figure 1 antibiotics-14-00146-f001:**
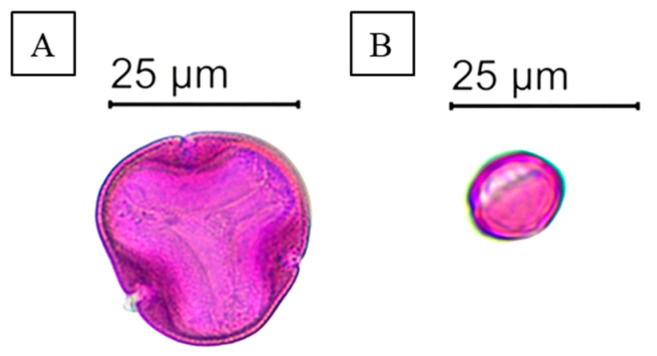
Microscopic pollen images of linden pollen (**A**) and chestnut pollen (**B**).

**Figure 2 antibiotics-14-00146-f002:**
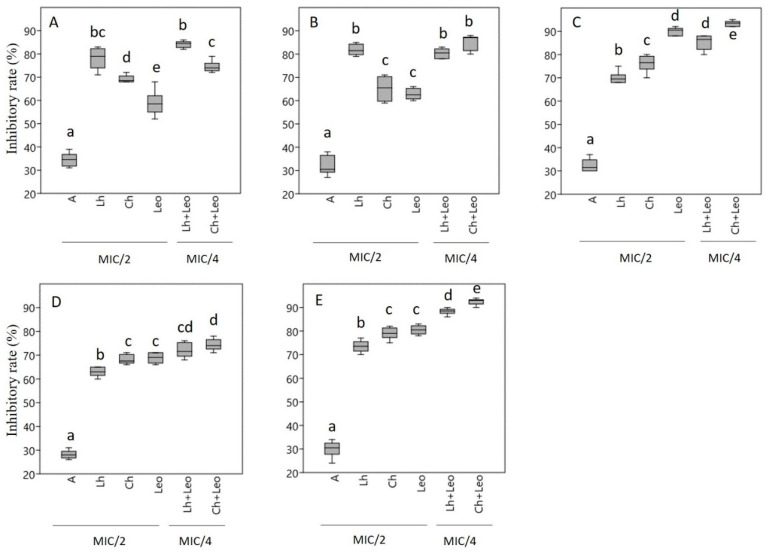
The biofilm degradation effects of linden and chestnut honeys and lavender essential oil alone and in combination against five bacteria. *H. influenzae* (**A**), *H. parainfluenzae* (**B**), *M. catarrhalis* (**C**), *P. aeruginosa* (**D**), and *S. pneumoniae* (**E**); antibiotic (A), linden honey (Lh), chestnut honey (Ch), lavender essential oil (Leo), combination of linden honey and lavender essential oil (Lh + Leo), combination of chestnut honey and lavender essential oil (Ch + Leo). Data were expressed using box plots with minimum to maximum values presented by vertical lines, with the median in the plot as a horizontal line. Different lowercase letters above the boxes indicate significant differences between the means of inhibitory rates of different components against the given bacteria, according to the Tukey’s test (*p* < 0.05), based on results of 6 parallel measurements (n = 6).

**Table 1 antibiotics-14-00146-t001:** Pollen spectrum of linden honey and chestnut honey.

Pollen Type—Relative Frequency (%)					
Honey Sample	*Tilia* spp.	*Robinia pseudoacacia*	*Brassica napus*	*Castanea sativa*	Other
Linden	36.0	25.7	20.4	6.2	11.7
Chestnut	0.2	4.3	1.4	93.1	1.0

**Table 2 antibiotics-14-00146-t002:** Sensory characteristics and physicochemical parameters of honey samples. Data are presented as means ± standard deviation (n = 3).

Honey Type, Plant Name	Sensory Characteristics (Color, Odor, and Consistency)	ABS_450–720_ (mAU)	Electrical Conductivity (mS/cm)	pH
Linden, *Tilia* spp.	Light amber, strong odor, semisolid, fine, granulated	107 ± 1.4	0.528 ± 0.01	4.1 ± 0.01
Chestnut, *Castanea sativa*	Dark amber, intense odor, semisolid, fine, granulated	215.3 ± 1.4	0.598 ± 0.01	4.7 ± 0.01

**Table 3 antibiotics-14-00146-t003:** Antibiotic sensitivity of test bacteria according to the disk diffusion assay.

Antibiotics	1	2	3	4	5
Amikacin	-	R	S	S	S
Ampicillin	R	R	S	R	R
Amoxicillin/ clavulanic acid	-	S	S	R	R
Ciprofloxacin	R	S	S	S	S
Ceftazidim	-	S	S	-	-
Eritromicin	-	-	R	-	S
Gentamicin	S	S	S	S	S
Imipenem	S	S	S	S	S
Levofloxacin	-	-	R	-	-
Oxacillin	R	-	-	-	-
Penicillin	R	-	-	-	-
Piperacillin/ tazobactam	-	S	S	-	-
Vankomicin	S	-	-	-	-

1: Moraxella catarrhalis, 2: Pseudomonas aeruginosa, 3: Streptococcus pneumoniae, 4: Haemophilus influenzae, 5: H. parainfluenzae; R: resistant, S: sensitive, “-”: no data.

**Table 4 antibiotics-14-00146-t004:** The minimum inhibitory concentrations (MICs) of linden and chestnut honey, lavender essential oil, and the antibiotic controls (amikacin for *H. influenzae* and *H. parainfluenzae*; imipenem for *M. catarrhalis* and *S. pneumoniae*; gentamicin for *P. aeruginosa*) against the investigated bacterial strains.

	MIC Values (mg/mL)				
	*S. pneumoniae*	*H. influenzae*	*H. parainfluenzae*	*M. catarrhalis*	*P. aeruginosa*
Linden honey	142.86	111.11	111.11	142.86	176.47
Chestnut honey	142.86	111.11	111.11	111.11	176.47
Lavender essential oil	0.31	0.63	0.63	1.25	2.50
Antibiotics	0.0008	0.0031	0.0016	0.0002	0.0063

**Table 5 antibiotics-14-00146-t005:** Combination MIC values of test samples (µg/mL).

Bacterial Strains	Test Samples	Combination MIC
*P. aeruginosa*	Leo	0.3125
	Lh	88.235
	Leo	0.3125
	Ch	44.1175
*S. pneumoniae*	Leo	0.0775
	Lh	71.43
	Leo	0.0775
	Ch	35.715

Leo: lavender essential oil; Lh: linden honey; Ch: chestnut honey.

**Table 6 antibiotics-14-00146-t006:** FICI values and combination effects of test samples.

Bacterial Strains	Test Samples	FICI	Combination Effect
*P. aeruginosa*	Leo	0.625	additive effect
	Lh		
	Leo	0.375	synergistic effect
	Ch		
*S. pneumoniae*	Leo	0.75	additive effect
	Lh		
	Leo	0.5	synergistic effect
	Ch		

Leo: lavender essential oil; Lh: linden honey; Ch: chestnut honey.

**Table 7 antibiotics-14-00146-t007:** Effect of chestnut honey, lavender essential oil, and their combination on the release of DNA in Gram-negative (*P. aeruginosa*) and Gram-positive (*S. pneumoniae*) bacteria.

DNA Release from Bacterial Cells (%)			
Samples	Concentrations	*P. aeruginosa*	*S. pneumoniae*
Chestnut honey (Ch)	MIC/2	11.7 ± 1.2	14.3 ± 1.3
	MIC	26.9 ± 2.3	31.6 ± 1.7
	MIC × 2	60.7 ± 2.5	66.3 ± 2.3
	MIC × 4	100	100
Lavender essential oil (Leo)	MIC/2	15.6 ± 1.8	18.1 ± 1.3
	MIC	36.5 ± 2.2	42.2 ± 2.1
	MIC × 2	67.1 ± 2.5	68.4 ± 2.3
	MIC × 4	100	100
Combination	MIC Ch + MIC Leo (1:1 ratio)	67.9 ± 2.8	69.8 ± 3.0

Data are means ± standard deviations of six independent determinations (n = 6).

**Table 8 antibiotics-14-00146-t008:** Kinetics of 260 nm absorbing material released from Gram-negative (*P. aeruginosa*) and Gram-positive (*S. pneumoniae*) bacteria treated with chestnut honey, lavender essential oil, and their combination.

DNA Release from Bacterial Cells (%)			
Samples	Time (min)	*P. aeruginosa*	*S. pneumoniae*
Chestnut honey (Ch) conc.: MIC × 2	0	0	0
	20	34.4 ± 2.1	39.2 ± 2.2
	40	55.2 ± 2.3	58.9 ± 2.1
	60	60.7 ± 2.5	66.3 ± 2.3
	90	66.6 ± 2.2	68.0 ± 2.6
Lavender essential oil (Leo) conc.: MIC × 2	0	0	0
	20	40.3 ± 2.2	49.1 ± 2.0
	40	62.8 ± 2.5	64.4 ± 2.6
	60	67.1 ± 2.5	68.4 ± 2.3
	90	69.2 ± 2.3	70.5 ± 2.8
Combination conc.: MIC Ch + MIC Leo (1:1 ratio)	0	0	0
	20	46.2 ± 2.4	53.6 ± 1.9
	40	63.7 ± 2.2	68.0 ± 2.1
	60	67.9 ± 2.8	69.8 ± 3.0
	90	70.9 ± 2.6	71.2 ± 2.2

Data are means ± standard deviations of six independent determinations (n = 6).

## Data Availability

The original contributions presented in this study are included in the article. Further inquiries can be directed to the corresponding author.

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
