# Peer review of "Synergistic Antibiofilm Effects of Chestnut and Linden Honey with Lavender Essential Oil Against Multidrug-Resistant Otitis Media Pathogens"

_antibiotics, 2025, doi:10.3390/antibiotics14020146_

Round 1

Reviewer 1 Report

Comments and Suggestions for Authors

The present work reports, for the first time, the anti-biofilm activity of linden or chestnut honey with an essential oil of lavender. Particularly, the authors delved into the potential of these natural products to tackle several otitis-causing bacteria. To do so, the authors employed techniques to ensure the botanical origin of the two tested honeys. Furthermore, the MIC and anti-biofilm potential of the tested honeys and essential oil were also determined alone and in combinations to search for synergistic effects. The assayed honeys and essential oil revealed low to mild MIC values. Regarding the inhibition of biofilm formation, the natural products were more promising than the tested antibiotics. However, the biofilm degradation assays were carried out with half of the MIC concentration, and these were considerably higher for the honeys and essential oil compared to the antibiotics. Ultimately, these natural products could be formulated as an auxiliary measure to fight bacterial ear infections, and in that sense, this article is very pertinent. Thus, I must congratulate the authors for this work. Nonetheless, I consider that this manuscript may be improved by addressing specific issues that I mention below.

Page 3, line 84 – Perhaps a more precise description would be to mention that those classes are linked to their biosynthetic pathways, instead of referring to them as belonging to different scaffolds. A possibility to improve this segment might be:” Essential oils (EOs) are mainly composed of volatile hydrocarbons and oxygen-containing compounds belonging to several biosynthetic classes. Terpenoids and phenylpropanoids are classes of naturally occurring compounds frequently found in EOs.”

Page 3, line 92 – Perhaps consider mentioning that all those compounds are monoterpenes.

Page 3, line 109 – Please consider using a more commonly used expression instead of “points of action”. I would suggest using modes of action or molecular mechanisms, for example. Also, please add a reference to this claim and rephrase it perhaps to sound less definitive. Overall, I would suggest, for example:” Honeys and essential oils might achieve their antibacterial effect through several modes of action, which in turn might hinder acquired bacterial resistance to these complex mixtures of compounds”

Page 3, line 119 – Please consider spelling Castanea spp. or Castanea sativa, since the species is mentioned later in the results section.

Page 3, line 124 – Please add that the essential oil was extracted though steam distillation to increase clarity.

Page 4, line 136 – In “Ten grams of honey were measured (…)” perhaps ”were” is more well suited.

Page 4, line 140 – Please change “10 ml to “10 mL”.

Page 4, line 146 – Perhaps the clarity of this phrase can be increase: “The pollen preparations were covered with a coverslip and analysed (…)”.

Page 4, line 159 – “(…) 20% honey solutions were prepared, (…)”. These 20% are w/w?

Page 6, line 233 – Could you please mention where the methanol was purchased from?

Page 6, line 245 – The statistical analyses need a revision, in my opinion. Student’s t-tests have assumptions that have to be checked before performing the tests. These include checking if there is normality of the differences, homogeneity of variance, and if the pairs of tests are independent from each other. Perhaps these assumptions were checked, but this has to be explicit stated in this section. If these assumptions are not met, the authors might use analogous non-parametric tests (e.g., Mann–Whitney U tests). Furthermore, since there are multiple t-tests, there is the need to control the risk of Type I error due to multiple comparisons. So, the p-values might be corrected with a Bonferroni correction, for instance. Also, refer to whether the tests are one- or two-sided. Lastly, it is possible to use a one-way ANOVA with Tukey's Honest Significant Difference (HSD) tests, for instance, instead of the corrected t-tests or Mann–Whitney U tests.

Page 7, line 254 – Please reflect if this is the best place to refer to these results matching the ones reported in the available literature. Perhaps the discussion would be a better suited section.

Page 7, line 255 – Please change “Tilia ssp.” to Tilia spp.”.

Page 8, line 286 – Please change “Tween 40” to “Tween40”, since this was the spelling adopted throughout this manuscript.

Pag 8, line 289 – Please consider adding the specific antibiotic used for each bacterium in the footer of Table 4.

Pag 11, line 325 – Consider rephrasing to demonstrate the importance directly, which can also simplify and clarify the message. For example: “Investigating the antibacterial and biofilm inhibition activities of natural substances is vital to explore new therapeutic options to curb the increasing spread of antibiotic resistance.” 

Pag 11, line 333 – I suggest that the authors rephrase this, possibly as follows: ”Both honey and essential oil have been hypothesized to achieve their antibacterial efficacy through several modes of action.”.

Pag 11, line 334 – Please consider rephrasing this part to perhaps increase its clarity: ”Essential oils are multicomponent, lipophilic liquids with characteristic aroma, as they are comprised of volatile compounds.”.

Pag 11, line 353 – Could you please include one or two references to support this phrase.

Pag 11, line 356 – Could you please include one or two references to support this phrase.

Page 12, 390 – Please consider changing “good” for “promising”, for instance, as not all honeys and essential oils might display extremely good bactericidal activities. 

Page 13, 419 – At line 416 it was spelled as “E. coli” and here as “Escherichia coli”. Please consider either switching the position of these two, or, since the genus Escherichia was referred to before, just keep “E. coli” for both cases.

Page 13, 427 – Please consider changing “S. Typhimurium” to “S. typhimurium”.

Page 13, 430 – I believe that the discussion is lacking a section where the downsides or fragilities of this study are mentioned. Furthermore, these might be connected with future prospectives which are also missing. Knowing what was reported here for the first time and the current literature, what would be the next steps? Perhaps conducting a chemical analysis of the honeys? Screening the bactericidal activity of individual main compounds of the essential oil tested? Testing other essential oils of the same genus or other Lamiaceae essential oils?

Author Response

Our responses to Reviewer 1 can be read in the attached file.

Reviewer 2 Report

Comments and Suggestions for Authors

The manuscript by Angyan et al presents a case for using linden or chestnut honey with lavender essential oils against otitis media causing MDR bacteria. No mechanism of action for the same is presented and the manuscript may benefit from this.

It is also not clear to me as to what makes this manuscript a scientific advancement as several papers mantioning antibacterial effects of essential oils and honey already exist. Merely mentioning that combining two substances enhances effects seems additive in nature. One can in principal also combine more number of substances and achieve better results, so what makes this study stand out is unclear.

Similarly all experiments have been done in invitro conditions on lab grown bacteria, I recommend using multiple disease models and validating the results there before considering publication.

In all it appears that this is a half baked story that needs to be further matured before any significant impacts can be considered.

Here are my specific comments:

The abstract is clear, concise and easy to understand, summarizing the research effectively. While the abstract mentions "bacteria causing otitis media," specifying the strains used (e.g., H. influenzae type b, S. pneumoniae serotypes) would enhance clarity and reproducibility. Mentioning the specific antibiotic(s)/groups used for comparison would strengthen the study's context. Including brief quantitative data may also be considered. Finally a brief on potential clinical implications might benefit the abstract as well.

Section 2.1: Be more specific about the "Hungarian apiary in the South Transdanubian region." If possible, provide a more precise location or the name of the apiary (if permissible).

Section 2.4: Briefly explain the rationale for measuring specific physicochemical parameters (e.g., "Electrical conductivity is a measure of the ionic content of honey").

Am I correct in assuming that honey solutions were prepared by dissolving honey in water? Would it make sense to mention this in the paper?

Section 2.8: If possible, specify the reason for using MIC/2 and MIC/4 concentrations for honey and essential oil, respectively, in the biofilm eradication assay.

Is MIC/2 or MIC/4 a standard term in the field, if not consider specifying/expanding/explaining it at the point of first use

Statistical Analysis: The authors may consider including information about the statistical assumptions underlying the t-tests and whether these assumptions were met.

If appropriate, consider using more advanced statistical analyses, such as ANOVA, to compare the effects of multiple treatments. To me it seems that all experiments were conducted using a single batch of honey and essential oils, is this true. I would recommend doing this with at least 3 batches and then reporting SD values for the experiments

Discussion of Results: Discuss the potential reasons for the observed differences in biofilm eradication activity among the different treatments.

Discuss the limitations of the study, such as the use of in vitro models and the limited number of bacterial strains tested.

Comments on the Quality of English Language

Its fine

Author Response

Our responses to Reviewer 2 can be found in the attached document.

Reviewer 3 Report

Comments and Suggestions for Authors

The manuscript is too general. The author should explore the underlying mechanism to prove their results.

Combining linden or chestnut honey with lavender essential oil 2 can eradicate biofilms formed by otitis media causing 3 multidrug-resistant bacteria

Comments about title: please consider changing title “Synergistic Antibiofilm Effects of Linden and Chestnut Honey with Lavender Essential Oil Against Multidrug-Resistant Otitis Media Pathogens

1.                 Comments:  Essential oils are composed of components such as Linalool, Linalyl Acetate, Camphor, 1,8-Cineole (Eucalyptol), Lavandulyl Acetate, Terpinen-4-ol, Borneol, Beta-caryophyllene, Geraniol, Coumarins, and Tannins. Most of these constituents exhibit various pharmacological properties. However, the study would have been more specific if the authors had investigated which component is primarily responsible for the synergistic antibiofilm activity.

Linden honey is rich in natural sugars such as fructose and glucose; essential vitamins like B1, B2, B6, and C; trace elements such as calcium, potassium, manganese, and zinc; flavonoids, polyphenols, and organic acids, which provide antioxidant and antimicrobial benefits. Additionally, it contains small amounts of enzymes like diastase and invertase that enhance its health-promoting qualities.

Chestnut honey, is rich in mineral content, including potassium, magnesium, iron, and manganese; tannins; phenolic acids; flavonoids, and antioxidants.

In my opinion, the study lacks clarity and fails to provide evidence on which specific component plays a synergistic role as an antibiofilm agent and which components are excessive or unnecessary, leading to potential wastage of resources. Not all constituents necessarily act as antibiofilm agents. It would have been more impactful if the authors had studied the main active constituent of each honey and designed a targeted therapy to eliminate bacteria within biofilms. This represents the primary flaw and drawback of the study, as it lacks depth and focus in its research approach.

References:

·        Shuai Zou, Heng Tao, Ya-Ning Chang; Characterization of antioxidant activity and analysis of phenolic acids and flavonoids in linden honey. Food Science and Technology. 2022. DOI: 10.1590/fst.76621

·        Dmitry V. Gruznov, Olga A. Gruznova, Alexey B. Sokhlikov, Anton V. Lobanov, and Irina P. Chesnokova. Changes in the chemical composition and antimicrobial activity of linden, buckwheat and sunflower honey stored at low temperatures. Current Research in Nutrition and Food Science

2.                 Please draw a roadmap to illustrate the study process, reflecting the key steps and methodologies undertaken.

3.                 Please include images of the biofilm structure before and after disruption through the application of each honey and essential oil.  Using confocal microscopy for this purpose would provide a detailed visual representation of the biofilm's formation and its subsequent disruption.

4.                 Please explain how the dose was selected of each antibiotic?

5.                 I recommend authors to elucidate the mechanism through which these honey and essential oil eradicate the biofilm.

Author Response

Our responses to Reviewer 3 can be read in the attached file.

Round 2

Reviewer 2 Report

Comments and Suggestions for Authors

Thanks a lot for providing a mechanism of action: Checkerboard assay is an interesting and relevant addition. However, the statement in Line 398 seems mistaken and needs to be corrected. In Table 7 I see data for Ch, Leo and Ch+Leo, what about Leo+Lh?? Can you explain why Ch+Leo at MIC DNA release is as good as independently from any one with no additive or synergistic effect.

Original comment “Similarly all experiments have been done in invitro conditions on lab grown bacteria, I recommend using multiple disease models and validating the results there before considering publication.” Has not yet been addressed. I continue to recommend using an actual disease model (atleast one) and then I would consider that the authors have nailed this study.

Finally the english in the manuscript still needs work to improve readablity.

Comments on the Quality of English Language

Needs improvements

Author Response

Comment 1: 

Thanks a lot for providing a mechanism of action: Checkerboard assay is an interesting and relevant addition. However, the statement in Line 398 seems mistaken and needs to be corrected.

Reply:

The statement was rephrased as follows:

“Based on the FICI values, the interactions between the test substances were determined, taking into account the following value ranges: FICI ≤ 0.5: synergist; 0.5-1: additive; >1: negative interaction; > 4: antagonist [61].”

Comment 2:

In Table 7 I see data for Ch, Leo and Ch+Leo, what about Leo+Lh?? Can you explain why Ch+Leo at MIC DNA release is as good as independently from any one with no additive or synergistic effect.

Reply:

Since preliminary studies confirmed that the Leo and Lh samples do not have a synergistic effect, membrane degradation studies were performed with Ch, Leo and Ch+Leo.

The first paragraph of section 3.6 in the revised manuscript contains the reasons why the studies were conducted with chestnut honey and lavender essential oil:

 “The checkerboard titration supported the synergistic effect of Ch and Leo (Table 6), so membrane degradation studies were performed with these two samples and their combinations. In addition, Ch was chosen for these experiments because of its high antibacterial and antibiofilm activity against S. pneumoniae (Gram-positive) and P. aeruginosa (Gram-negative) bacteria (Table 4, Figure 2).”

Comment 3:

Original comment “Similarly all experiments have been done in invitro conditions on lab grown bacteria, I recommend using multiple disease models and validating the results there before considering publication.” Has not yet been addressed. I continue to recommend using an actual disease model (atleast one) and then I would consider that the authors have nailed this study.

Reply:

Since we do not have a study that explores the activity of the combined effect of lavender essential oil and varietal honeys, in our opinion the tests carried out against the most common pathogens underlying the development of otitis media provide sufficient information value in the manuscript. This allows us to model the effectiveness of the samples and their combinations.

For this reason, the authors do not feel justified including another strain in the study.

Comment 4:

Finally the english in the manuscript still needs work to improve readablity.

Reply:

We have involved a native speaker of English to check language issues in our manuscript.

Reviewer 3 Report

Comments and Suggestions for Authors

The authors have provided the underlying mechanism of the bactericidal activity of the synergistic antibiofilm effects of chestnut linden or linden chestnut honey with lavender essential oil. This was my main concern and the primary drawback of this study. However, I am pleased that the authors have addressed this issue very effectively. As a result, I recommend the acceptance of the manuscript for publication

Author Response

Comment:

The authors have provided the underlying mechanism of the bactericidal activity of the synergistic antibiofilm effects of chestnut linden or linden chestnut honey with lavender essential oil. This was my main concern and the primary drawback of this study. However, I am pleased that the authors have addressed this issue very effectively. As a result, I recommend the acceptance of the manuscript for publication.

Response:

We are grateful to the reviewer for appreciating our efforts to improve the manuscript.